# Can Stimulus Valence Modulate Task-Switching Ability? A Pilot Study on Primary School Children

**DOI:** 10.3390/ijerph19116409

**Published:** 2022-05-25

**Authors:** Giulia D’Aurizio, Daniela Tempesta, Gennaro Saporito, Francesca Pistoia, Valentina Socci, Laura Mandolesi, Giuseppe Curcio

**Affiliations:** 1Department of Biotechnological and Applied Clinical Sciences, University of L’Aquila, 67100 L’Aquila, Italy; giulia.daurizio@univaq.it (G.D.); daniela.tempesta@univaq.it (D.T.); gennaro.saporito@graduate.univaq.it (G.S.); francesca.pistoia@univaq.it (F.P.); valentina.socci@univaq.it (V.S.); 2Department of Humanities, University Federico II, 80138 Napoli, Italy; laura.mandolesi@unina.it

**Keywords:** executive function, emotions, attention, frontal lobe, cognitive development

## Abstract

Executive functions and emotional processes follow a time-dependent development that reflects the brain’s anatomo-functional maturation. Though the assessment of these cognitive functions is largely examined, in children the role of emotions in the mental set-shifting is still rarely investigated. The aim of this study was to assess how attention shifting can be modulated by the valence of emotional stimuli. To this end, sixty-two primary school children were tested with a new emotional task-switching paradigm obtained by manipulating the emotional valence and physical features of the stimulus pool. Thus, two tasks were alternatively presented: the Valence task and the Color task. Based on executive performance results, we found a lengthening of response times and a lower accuracy in the emotionally connoted task (Valence task), compared to the neutral task (Color task). The data demonstrate that the processing of emotional stimuli modulates the task-switching performance during development. These findings could help in the implementation of teaching strategies that can promote the development of executive functions and, therefore, functionally improve the overall academic performance of children. Finally, a better understanding of the developmental trajectories of executive functions can help neuropsychologists both in the early diagnosis and treatment of potential executive alterations.

## 1. Introduction

Executive functions (EFs) modulate human behavior by making the individual able to cope with requests from an ever-changing external environment. In this sense, the processes of cognitive and behavioral flexibility, problem-solving, action monitoring and sequence processing, feedback analysis, and processing skills, all act in order to correct and reprogram the mental set. These functions have strong adaptive meaning in the processes of executive control, the engagement/disengagement of attention, and mental set shifting [1].

According to the model proposed by Miyake and colleagues [2], EFs modulate cognitive functioning through the three processes of shifting, working memory and inhibition. In this theoretical framework, shifting refers to the mental set-shifting process. It allows shifting between different tasks and, therefore, mental-set reconfiguration following environmental-feedback analysis [3]. Furthermore, shifting modulates both cognitive and behavioral flexibility [4]. Working memory refers to the updating, monitoring and manipulation of information necessary to other cognitive functions [5]. Inhibition refers to voluntary suppression of dominant and automatic responses to favor a new goal-directed behavior [6]. This process is modulated by selective attention [7].

Zelazo and Muller [8] split EFs into two components: hot and cold, respectively. Hot EFs are related to the emotional aspect of cognition, cold EFs include purely cognitive components. More specifically, hot EFs involve emotion regulation, reward processing risky decision making, social cognition, and affective decision, whereas cold EFs include response inhibition, working memory, executive control, cognitive flexibility, mental-set shifting, and attention [9]. Based on a network-driven approach, Salehinejad and colleagues [10] proposed “a prefrontal-cingular network that can explain neuronal correlates of hot versus cold EFs more comprehensively”. In this network, the orbital and medial prefrontal cortex (PFC), and ventral anterior cingulate cortex (ACC) are more involved in the hot component of EFs, whereas the lateral PFC and dorsal ACC are involved in cold EFs.

In school-age children, EFs mediate and promote the development of the self-regulation processes that increase the probability of adaptation to the environment through the emergence of skills such as online monitoring of behavior during the execution of an action, verification of the objectives achieved, comparison with the expected results and, therefore, possible redefinition of the same in case of discrepancy.

It is also thanks to development of these cognitive processes that they will progressively become more able to recognize their emotions and the consequent behaviors they elicit [11,12]. This ability, therefore, becomes fundamental in the modulation of interaction with others within the school context, in which there will be a continuous strengthening of executive functionality and, consequently, an improvement in social and interaction skills [13,14] in a reverberant circuit in which the former would favor the latter and vice versa.

Similarly to other cognitive processes, EFs also follow a time-dependent development that reflects, beyond individual differences, the processes of a general anatomo-functional brain maturation and a development of specific areas directly involved in these functions (i.e., frontal and prefrontal areas; [15,16,17]). The correct and complete development of these areas will therefore mark the acquisition of a given function within a specific time window and will inevitably be modulated by genetic and environmental factors [18,19].

Neuroimaging studies conducted on children and adolescents showed an increase in prefrontal and frontal-subcortical circuit activation during the execution of paradigms stressing executive functioning [20,21], as well as the presence of patterns of decreased neural activation in the same areas [22]. It would be reasonable to find this discrepancy in the intrinsic characteristics of the maturation processes and would favor the intervention of age-related mechanisms capable of regulating and setting neural activity, so that it can pass from more widespread activation patterns, progressively, to more focal patterns [23]. This process would result in a gradual differentiation of the resulting executive processes which would emerge secondary to the mechanisms of improvement of neural activity within these networks.

Furthermore, numerous studies have shown that, in children, the functional complexity of executive processes is supported by extensive neural circuits that involve not only PFC, but also the projections it establishes with the limbic subcortical areas involved in the emotional regulation processes [24,25,26], in the analysis of the emotional stimulus and in the behavior consequently elicited [27,28].

Jurado and Roselli [29], based on the concept of a “time window” of development, identify sensitive periods within which the anatomo-functional development would allow the sequential acquisition of executive functions, underlining how, for example, between the sixth and tenth year of life can be observed the acquisition of the ability to inhibit behavior and its planning and mental set-shifting: such executive abilities are typically involved in the processes of executive behavior control and attentional switching.

Typically, one of the most widely used paradigms for the study and measurement of mental set-shifting processes is task-switching [30,31]. The processes of executive attention and attentional switching account for the process of disengagement of attention from a stimulus and the reallocating of it toward a new stimulus, for the purpose of implementing an adaptive and goal-oriented behavior. To modulate this process and that of cognitive flexibility, numerous factors would have to intervene at different levels, including emotions [32,33].

The aim of the present study was to understand whether and how both the executive control of behavior and the consequent processes of inhibition and attention shifting [11] can be modulated by the valence of a set of emotional stimuli in primary school children. Starting from a standard task-switching (TS) protocol [34], we created a new task with the aim of emphasizing the emotional component of the stimulus. The experimental manipulation involves the emotional stimulus-pool, the interval stimulus-response and, finally, the exposure time both at the target and the cue.

## 2. Materials and Methods

### 2.1. Participants

We recruited 62 students from the fourth and fifth grades of primary school (28 M; mean age: 9.77 ± 0.42). Children with a certified diagnosis of specific developmental disorders or with other clinical conditions were excluded from the study.

Furthermore, in order to exclude a possible impairment in the process of recognition of primary emotions or by a not-yet-complete acquisition of this ability, participants were administered the Test of Emotion Comprehension (TEC) [35,36] regarding the two components of recognition and external cause. Specifically, the recognition component refers to the ability to recognize the facial expressions of primary emotions, whereas the external cause component identifies the ability to understand how external factors can elicit and modulate one’s own emotions and those of others. Table 1 shows the demographic characteristics and TEC scores of the participants in the study.

After obtaining the authorization of the school director, and before the experimental stage, the researchers involved in the study presented the experimental protocol to the parents, explaining the main aims of the study and the cognitive tasks to be administered to the children. At this point, on a voluntary basis, informed consent from the parents was acquired. Only after these steps did data collection start. Moreover, ethical approval for the present study was provided by the Internal Review Board of the University of L’Aquila (n.16/2016).

### 2.2. Procedure

All participants were tested in a sound-proof and well-illuminated room within the school building. After the TEC assessment, children started the computer-based administration of emotional task-switching: to this end, a laptop of 15′′ was used and children were positioned at 50 cm from the screen; answers were done by tapping on specific keys on the keyboard (see above).

#### Emotional Task-Switching Protocol

Usually, standard task-switching [37,38,39,40] involves two tasks presented in rapid and randomized sequences in order to obtain both the switch (SW-t) and repetition trials (REP-t) depending on alternation or repetition between the two different tasks. Thus, the task allows measuring the participants’ response times and accuracy of the switch and repetition trials, as well as of the switch cost (SC). The SC can be defined as the cognitive cost underlying the process of engagement/disengagement attention and the reconfiguration of the mental set.

Based on this protocol, we developed an ad-hoc emotional task-switching by manipulating parameters of emotional valence and physical features (see Figure 1). Thus, in the present paradigm, two tasks were alternatively presented: (1) The Valence task (tA), in which eight different cartoon faces expressed two basic emotions (i.e., happiness and anger) with opposite valences. A cue-emoticon indicated the task to be performed. (2) The Color task (tB), in which the same stimulus-set of Valence task was presented in the two different conditions (i.e., black-white and color). A cue-palette indicated the task to be performed.

In each task, the two different conditions (anger/happiness versus black-white/colored) were balanced so that the participant was exposed the same number of times to each of them. The emotional task-switching consisted of both a practice session (1 block of 80 trials) and an experimental session (4 blocks of 80 trials each). With respect to the event’s trial-timing, a cue (emoticon or palette) was presented for 1000 ms, and then it was followed by a target that stayed on the monitor for 3000 ms, or until the participant responded. Moreover, for both tasks the same two response keys were provided and marked on the keyboard by a red (S, left index finger) and a yellow square (L, right index finger), respectively. An illustration of the events timing in the emotional task-switching is illustrated in Figure 2.

Stimuli presentation and response recordings were managed by means of Superlab software (version 4.0.2 for Windows, Cedrus Corporation, San Pedro, CA, USA).

### 2.3. Statistical Analyses

In order to understand how executive performance varied with respect to both the trial type (REP-t versus SW-t) and the task (Valence versus Color), for the global emotional TS performance (Model 1, Global performance) and for each of the two different tasks (Models 2 and 3, respectively, Valence and Color), the following dependent variables were computed: (1) median reaction times (in ms; median RT) to switch and repetition trials, (2) switch costs (SCs, in ms, were computed as the difference between mean switch RT and mean repetition RT), and (3) angular transformations of the proportion of errors. For each subject, proportions of errors were computed by including both incorrect and missing responses. Moreover, it was submitted to an angular transformation, y = arcsen [sqr(p)], where sqr(p) is the square root of the proportion.

All TS dependent variables, except SC, were submitted to a mixed-model analysis of variance (ANOVA).

In Model 1 both Trial (SW-t versus REP-t) and Task (Valence versus Color) are considered within-factors. In Models 2 and 3 Valence (positive versus negative) and Saturation (black-white versus color) were also entered as within-factors.

Regarding SC, in order to assess the potential difference both in global emotional task-switching performance and within each task (tA and tB), a one way-ANOVA was carried out on the two conditions: positive versus negative (for the Valence factor) and black-white versus color (for the Color factor).

Alpha level was fixed to ≤0.05 and all statistical analyses were performed using IBM SPSS Statistics for Macintosh, version 25.0 (IBM Corp., Armonk, NY, USA).

## 3. Results

### 3.1. Model 1: Emotional Task-Switching Global Performance

Reaction times: A significant main effect showed both for the task (F_1.61_ = 53.8; *p* < 0.0001; ηp^2^ = 0.47; Figure 3A) and trial (F_1.61_ = 53.8; *p* < 0.001; ηp^2^ = 0.43; Figure 3B), indicating a slower RT in Valence task (1221.67 ± SE 37.33) with respect to the Color task (1046.51 ± SE 34).

As expected, the result showed a slower RT in switch trials (1188.01 ± SE 37.81) with respect to repetition trials (1080.08 ± SE 30.97).

No other main effects or interactions were statistically significant.

Error proportion. A significant main effect was found for trial (F_1.61_ = 54.51; *p* < 0.001; ηp^2^ = 0.47; Figure 4), indicating a lower accuracy in switch (1.21 ± SE 0.2) with respect to repetition trials (1.28 ± SE 0.21). Moreover, a trend towards significance was found for the task (F_1.61_ = 3.82; *p* = 0.05; ηp^2^ = 0.06; Figure 4B) showing a better performance in the Color (1.28 ± SE 0.25) with respect to Valence task (1.22 ± SE 0.26).

Finally, the ANOVA showed a significant task–trial interaction (F_1.61_ = 4.45; *p* = 0.04; ηp^2^ = 0.07) indicating a higher accuracy when the subject carried out the Color task and the repetition trials (Color REP-t = 1.32 ± SE 0.02; color SW-t = 1.24 ± SE 0.05; valence REP-t = 1.25 ± SE 0.026; valence SW-t = 1.2 ± SE 0.03).

### 3.2. Models 2 and 3: Valence and Color Task-Switching Performance

#### 3.2.1. Valence Task

Reaction times: A significant main effect was found for the trial (F_1.61_ = 19.75; *p* < 0.001; ηp^2^ = 0.24; Figure 5A) evidencing, as expected, that RTs were faster in repetition (1183.26 ± SE 33.67) than in switch trials (1270 ± SE 42.8).

Error proportion: A significant main effect emerged both for the trial (F_1.61_ = 19.04; *p* < 0.001; ηp^2^ = 0.24; Figure 6A) and for valence (F_1,61_ = 23.36; *p* < 0.001; ηp^2^ = 0.28; Figure 6B), bringing out a higher accuracy in the repetition (1.26 ± SE 0.028) than switch trials (1.2 ± SE 0.028) and for positive (1.6 ± SE 0.02) than negative stimuli (1.45 ± SE 0.03).

#### 3.2.2. Color Task

Reaction times: A significant main effect was found for the trial (F_1,61_ = 69.32; *p* < 0.001; ηp^2^ = 0.53; Figure 5B), indicating that RTs on repetition (990.72 ± SE 34) were faster than those on switch trials (1122.57 ± SE 35.34).

No other main effects or interactions were statistically significant.

Switch cost: A significant main effect was seen for the task (F_1.122_ = 4.9; *p* < 0.03; ηp^2^ = 0.04; Figure 7): a lower SC resulted from the Valence task (79.64 ± SE 14.6) than the Color (124.2 ± SE 13.81) task.

## 4. Discussion

The results demonstrate how task-switching is sensitive to the measurement of mental set-shifting and executive control processes in behavior during development and, in particular, in the age group of the participants. These data help to confirm some previous studies, conducted both on adults [41,42,43] and, to a lesser extent, on children [18,35], showing a decrease in response times and an increase in accuracy in sequences where there is no alternation between tasks (repetition), compared to when this experimental condition (switch) is instead present.

These results are in line with a previous study showing an age-dependent gradient of attentional processes: between 6 and 8 years of age there is a development of both executive control and response inhibition which tends to stabilize around 8 years of age [44]. Moreover, executive control, response inhibition, and set-shifting have been investigated from an ontogenetic perspective. The results confirm that the first EF to emerge is the response inhibition followed by selective attention and mental set-shifting [29].

Moreover, what is interesting is that the emotional processes, specifically the valence of the emotional stimulus, would also intervene to modulate this executive process. The data presented so far, in fact, describe a lengthening of response times and a lower accuracy that characterizes, in general, the performance of the participants in the execution of the emotionally connoted task (Valence task), compared to the neutral task (Color task).

This characteristic is strengthened by the results obtained by comparing the performance shown in the presence of emotional stimuli of opposite valence: a less accurate performance, in fact, characterizes the responses provided to stimuli with a negative value, compared to those with a positive value.

During development, children are continuously exposed to emotional contexts, particularly within social interactions (i.e., at school). In this context, a better environment adaptation process requires learning of the executive ability to switch from different emotional stimuli with different positive, negative and neutral valence. According to Martins and colleagues [45], “the ability to alternate between processing emotional and non-emotional information” is defined as affective flexibility.

Typically, basic mechanisms that ensure the implementation of fast and adaptive behavioral patterns intervene to modulate the processing of negatively connoted emotional stimuli [25,46,47]. In this sense, a decrease in accuracy would be justified by the need to respond immediately to input from the external environment. The data described above would therefore be interpretable, in light of this adaptive compensation mechanism, by the promptness of response with respect to accuracy and would add a piece to what is present in the literature regarding the existence, instead, of an opposite effect with respect to what emerged from our results [48].

The interaction between the executive control of behavior, the attentional processes, and the mechanism for regulating the emotional process ensures that goal-directed adaptive behavior can be programmed and executed. This interaction is also confirmed by neuroimaging studies which show that during executive control and emotional process regulation, there is a large overlap of neural activation in the CPF, amygdala and insula [49,50,51,52].

With the aim of investigating the task-switching abilities in children at the end of their primary school years, and to understand if and how the use of emotionally connoted material within the task elicited phenomena related to response times and accuracy (different from those traditionally present in task-switching paradigms), an ad hoc protocol was developed, which takes into account the value of the stimuli. For this purpose, images were chosen that were as familiar as possible to the recipients of the study and which considered the skills, emotional development, and executive processes of the specific age group of our sample.

In our opinion, taking into account the recently increased interest in the assessment of EFs during childhood, our findings may have significant future implications for clinical and scientific advantages. Firstly, investigating how EF, and in particular the engagement/disengagement processes of attention, mental set-shifting, behavior inhibition, and the mechanisms by which the processing of emotional stimuli modulate behavior during development, could contribute to the definition of didactic interventions. These interventions could favor the implementation of educational strategies that, taking into account the need to promote and favor the development of executive functions, could functionally improve the overall academic performance of children [28,53,54]. Parallel to the implementation of innovative didactic training, a better understanding of the process underlying the correct development of executive functions could contribute to advances in identifying increasingly sensitive and ecologically neuropsychological assessment tools in the early diagnosis of executive dysfunction. Executive functioning plays a pivotal role both in social ability development and school performance. Conversely, their dysfunction can cause psychopathological disorders and behavioral alterations.

Finally, continuing to investigate these cognitive processes and their development trajectories remains a scientific priority to promote correct neurocognitive function and to stimulate better cognitive, social and emotional development.

## 5. Conclusions

During development ontogenic processes of neural and cognitive maturation define specific patterns of executive task-switching functionality. In the task-switching used, by manipulating both the emotional valence and physical features of the stimulus pool, we found an increase in the response times and a lower accuracy in the emotional task compared to the neutral task. Nonetheless, the present study has some limitations. The first is related to the limited sample size, which needs to be increased in future studies. The second is related to the interest in tracing an evolutionary line of these functions, testing samples of children belonging to different classes from primary school to secondary school. This would surely allow researchers to follow the ontogenetic trajectory of these subjects, thereby having comparative data to be used as a reference when studying patients with atypical development. As a third limitation, it should be noted that the study did not consider the role that some factors, such as family context, attachment style, and quality of social relations children in the peer group, may have in executive function development. Future studies should consider these factors as possible mediators.

The present findings demonstrate that emotional stimulus processing can modulate task-switching performance during development and could help in the definition of educational strategies able to improve children’s academic performance. Parallel to the implementation of educational interventions, our results could support the development of screening and neuropsychological assessment tools.

## Figures and Tables

**Figure 1 ijerph-19-06409-f001:**
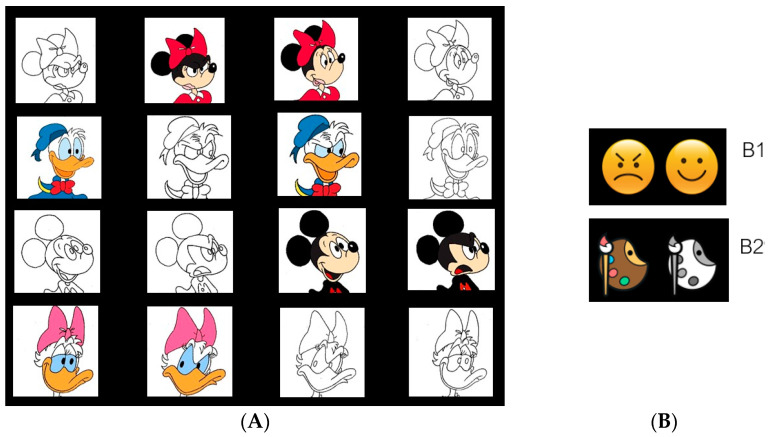
Stimuli (**A**) and cue (**B**) used in the emotional task-switching (note: B1 = cue used in the Valence task; B2 = cue used in the Color task). The stimuli were modified.

**Figure 2 ijerph-19-06409-f002:**
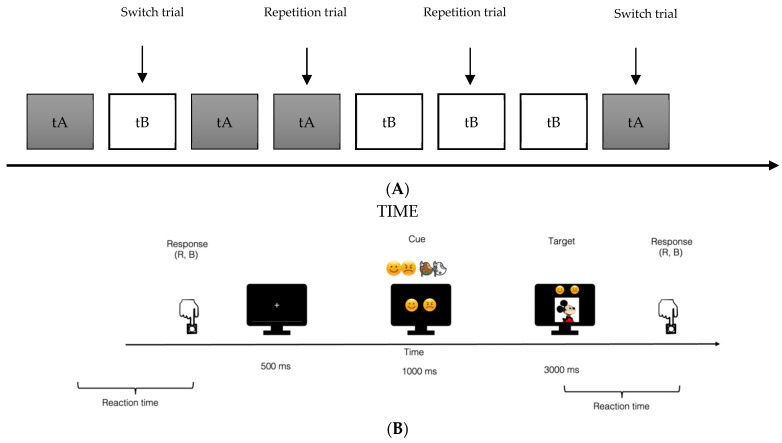
Emotional task-switching (**A**) and timing of events (**B**). tA = Valence task (“it’s happy or angry?”); tB = Color task (“it’s in black & white or coloured?”).

**Figure 3 ijerph-19-06409-f003:**
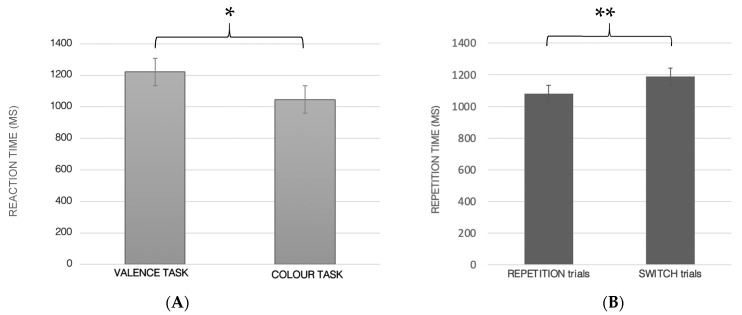
Reaction time (in MS) in the Valence and Color task (**A**); reaction time in the repetition and switch trials (**B**); * *p* < 0.0001, ** *p* < 0.001.

**Figure 4 ijerph-19-06409-f004:**
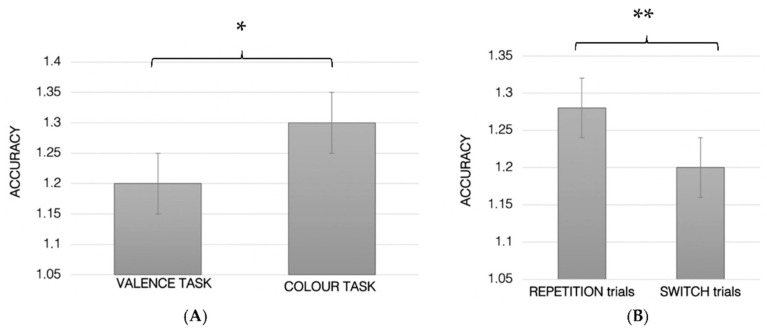
Accuracy in the Valence and Color task (**A**); reaction time in the repetition and switch trials (**B**); * *p* = 0.05, ** *p* < 0.001.

**Figure 5 ijerph-19-06409-f005:**
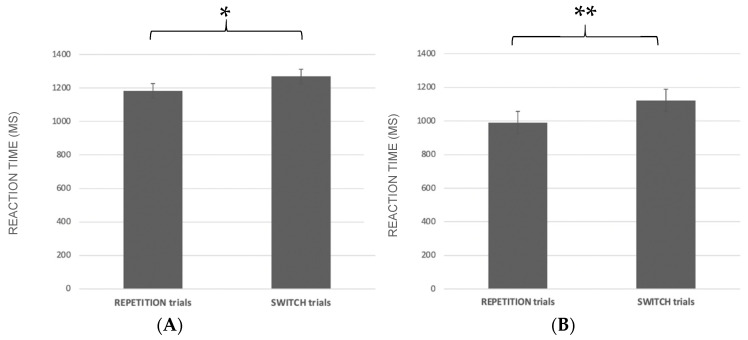
Reaction time (in ms) in the Valence and Color task (**A**); reaction time in the repetition and switch trials (**B**); * *p* < 0.001, ** *p* < 0.001.

**Figure 6 ijerph-19-06409-f006:**
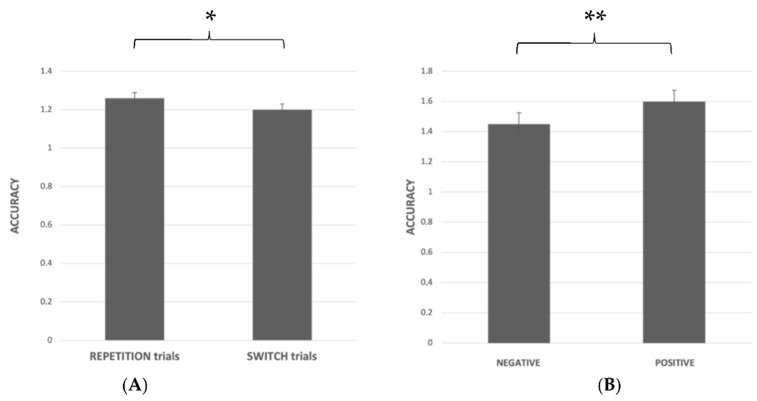
Valence task accuracy in repetition and switch trials (**A**) and in the negative and positive stimuli (**B**); * *p* < 0.001, ** *p* < 0.001.

**Figure 7 ijerph-19-06409-f007:**
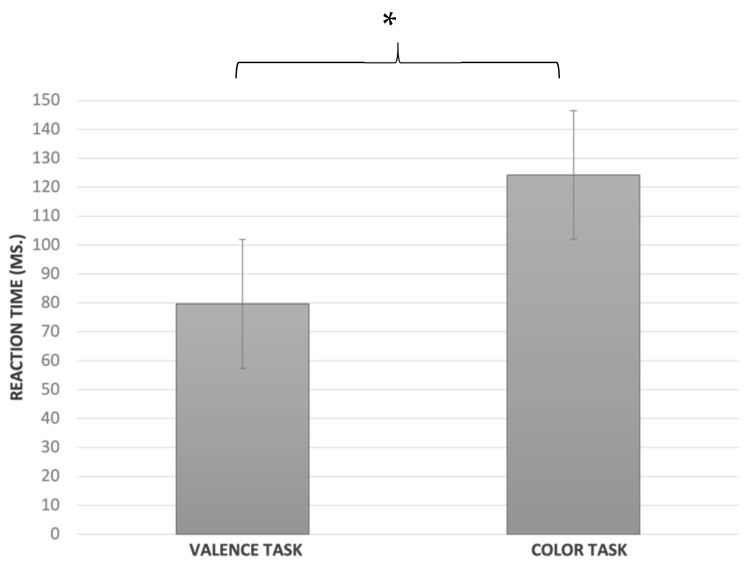
Switch cost in the Valence and Color tasks; * *p* = 0.028.

**Table 1 ijerph-19-06409-t001:** Demographic information (mean ± standard deviation) and TEC scores on the investigated sample.

	Females (*n* = 34)	Males (*n* = 28)	*p*
Age (mean ± SD)	9.76 ± 0.43	9.78 ± 0.41	n.s.
Education (mean ± SD)	4.6 ± 0.48	4.57 ± 0.5	n.s.
TEC	2	2	n.s.

TEC = Test of Emotion Comprehension; n.s. = not significant.

## Data Availability

Data is contained within the article.

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
