# Peer review of "Can Stimulus Valence Modulate Task-Switching Ability? A Pilot Study on Primary School Children"

_ijerph, 2022, doi:10.3390/ijerph19116409_

Round 1

Reviewer 1 Report

The idea of this study is well-motivated and clear. I hope my comments will be useful both for Authors and Editorial team, though I am not a native speaking reviewer. I agree there are not many systematical research on how emotional valence of stimuli can modify the scores obtained in task switching tasks. Usually  they engage „cold” executive functions and are useful  for assessment of mental flexibility and interference control based on data which have no specific connotation for children. However, especially in children, using emotional versus neutral matherial can significanly change the results as different motivational processes and different type of mental conflict may occur during performance.

Article has many merits. However, below I would like to outline some points which, in my opinion, could be taken into further confideration before publication:

In abstract section I would outline that „(..)this knowledge is important for teachers..”, but also for diagnosticians.

Introduction section is a good synthetic basis of the research, however I think Author/s could explain better:

1)the mechanism of mediation of EF on self-regulatory processes. How exactly does that happen? It would be good to give a short explanation, two or three sentences with examples of such mediation.

2) the heterogenous structure of EF (shortly). I would suggest to give examples of cold and hot executive functions, especially because it refers closely to the problem raised in this research.

In Matherial section: what numer 2 refers to in Table 2?

Limitations of this research could be combined with conclusions as these two separate parts are pretty short.

Author Response

Response to Reviewer 1 Comments

Point 1: In abstract section I would outline that „(..)this knowledge is important for teachers..”, but also for diagnosticians.

Response 1: This suggestion has been accepted. In the current version of the manuscript, we have highlighted this aspect both in the Abstract and in Discussion sections.

Point 2: The mechanism of mediation of EF on self-regulatory processes. How exactly does that happen? It would be good to give a short explanation, two or three sentences with examples of such mediation.

Response 2: We thank the Reviewer for this comment. In the current version of the manuscript, we have clarified and explained this concept in the Introduction section.

Point 3: The heterogenous structure of EF (shortly). I would suggest to give examples of cold and hot executive functions, especially because it refers closely to the problem raised in this research.

Response 3: We thank the Reviewer for the opportunity to clarify these issues. In the Introduction of the current version of the manuscript, we have included more details to explain executive functioning models, including the distinction between hot and cold EFs.

Point 4: In Matherial section: what numer 2 refers to in Table 2?

Response 4: In Table 2, number 2 refers to the participants’ TEC score obtained on the two Recognition and External Cause components. This is the highest score that can be obtained for the two TEC subscales. In the current version of the Table 2 we explained the meaning of such value.

Point 5: Limitations of this research could be combined with conclusions as these two separate parts are pretty short.

Response 5: We agree with the Reviewer, but such brief conclusions were made in accordance with the journal's guidelines. Nonetheless, in the current manuscript version, we added the limitations of the study in the Conclusion section.

Reviewer 2 Report

I would like to thank the authors for their work and effort. This is an interesting manuscript that addresses current and important issues. However, there are some considerations I must make.

1.- The introduction is very short. It deals with widely studied variables such as emotions and executive functions, therefore, this section should be expanded with background and different theoretical models.

2.- In the method and materials, the participation and ethics procedure is not described. Was an ethics protocol followed? What body approved it? Was informed consent given? All this should be described. The authors report the Helsinki declaration and the approval of the university, but was any national or regional ethics body or department involved? How was informed consent given to the families? Were explanatory meetings held?

The statistical analysis and results are well presented.

4.- The discussion is very short, as is the introduction. By expanding the first, the second will necessarily be expanded.

5.- Other aspects such as the limitations of the study itself and its future prospects should be dealt with. In addition, what are the theoretical and practical contributions of the manuscript and its findings? This should be clearly stated by the authors in their conclusions.

I encourage authors to address these questions to improve the quality of their manuscript.

Author Response

Response to Reviewer 2 Comments

Point 1: The introduction is very short. It deals with widely studied variables such as emotions and executive functions, therefore, this section should be expanded with background and different theoretical models.

Response 1: As requested also by Rev#1 we extended the Introduction by including more details to explain executive functioning models and the distinction between hot and cold EFs.

Point 2: In the method and materials, the participation and ethics procedure is not described. Was an ethics protocol followed? What body approved it? Was informed consent given? All this should be described. The authors report the Helsinki declaration and the approval of the university, but was any national or regional ethics body or department involved? How was informed consent given to the families? Were explanatory meetings held?

Response 2: In the revised version of the manuscript, we have clarified these aspects of the research. Briefly, the experimental protocol obtained both the approval of the university's ethics committee and the school Director. In an explanatory meeting, the researchers illustrated the experiment protocol to the parents, describing the type of study, the main aim, and the cognitive task that would be administered to the children. Then, on a voluntary basis, informed consent from the parents was acquired.

Point 3: The statistical analysis and results are well presented.

Response 3: We thank the Reviewer for this positive comment.

Point 4: The discussion is very short, as is the introduction. By expanding the first, the second will necessarily be expanded.

Response 4: As said also in point #1 the revised version of the manuscript, we expanded both Introduction and Discussion.

Point 5: Other aspects such as the limitations of the study itself and its future prospects should be dealt with. In addition, what are the theoretical and practical contributions of the manuscript and its findings? This should be clearly stated by the authors in their conclusions.

Response 5:  The revised version of the manuscript, in the final portion of the Introduction, includes the possible future implications of our results. Moreover, in the Conclusion, we also better explained the limitations of the study.

Round 2

Reviewer 2 Report

Thanks to the authors for addressing all the proposed improvements. The manuscript is now much clearer than at the previous stage and reports on important aspects that were missing.